# Was a Confessional Agreement in Early Modern Europe Possible? On the Role of the Sandomir Consensus in the European Debates

Maciej Ptaszyński

Faculty of History, University of Warsaw, PL 00-927 Warsaw, Poland; m.ptaszynski@uw.edu.pl

**Abstract:** The purpose of this article is to present the continuity of irenic thought in early modern times using the example of a confessional agreement concluded in 1570 in Poland, called the *Sandomir Consensus* (*Consensus of Sandomierz*). The initiators and authors of the document were Calvinists at the time, but the document's theologians soon attributed it to the post-Hussite community of the Bohemian Brethren. Here, the point of departure is the 1605 publication of the history of the Bohemian Brethren in Heidelberg, with an appendix containing the *Consensus*. The article explains the meandering origins of this historical interpretation: its roots in confessional polemics, as well as its legacy, to prove that irenicism was not a marginal or apolitical movement but actively contributed to shaping modern attitudes toward religion.

**Keywords:** irenicism; Protestantism; early modern history

## 1. Introduction

In 1605, a printer in Heidelberg published a history of the Bohemian Brethren, a minor branch of the Hussites movement (Camerarius 1605). As the author of the work, the title page named Joachim Camerarius, a famous humanist who had died over thirty years earlier (1574). The book was prepared for print by a cousin of this humanist, who also added an appendix with thirteen historical sources supporting the narrative. The last source added was a document known as the *Sandomir Consensus* (*Consensus of Sandomierz*), the inter-Protestant agreement reached in Poland in 1570.

The work raises some substantial questions: Why was the senior Camerarius so interested in the Bohemian Brethren that he composed its history? Why did the junior Camerarius publish a treatise that had emerged a generation earlier? Why did he reprint the *Sandomir Consensus*—a document that had originated in Poland, not Bohemia—at the conclusion of his treatise on Bohemian history? How did the meaning of the two documents change over the years? Did the book influence historiography, politics, or religious debate?

These questions should lead to the central issue of this article: the history of irenicism and its impact on the formation of state–church relations in the modern era. A dominant viewpoint in modern historiography presents the demand for the separation of state and church as a response to the search for religious peace, so strongly advocated for by John Locke and the British Enlightenment, an interpretation that William Cavanaugh has called "the myth of wars of religion" (Cavanaugh 2009, pp. 121–80; Gregory 2012, pp. 129–79). In this vision, in the course of the religious wars, peace, order, and discipline could only prevail as a result of state interference, as well as the progress of secularization (Taylor 2007, pp. 111–30). The place of the quarrelsome theologians had to be taken by the pragmatic "les politiques," and political theory replaced theology (Salmon 1959; Schnur 1962, pp. 9–11; Holt 1986; Kosseleck [1959] 1998, pp. 17–22; Beame 1993; Laursen and Villaverde 2012, p. 3; Daussy 2014; Forst 2013, pp. 138–69). In other words, peace and order could not be born from within the maelstrom of conflicting confessional churches, which were unable to break the paradigm of a monopoly on truth. Following this line of argumentation, some

historians claimed that only rules of coexistence ("practice of toleration," "coexistences confessionelles") could establish durable peace in Europe, while the reconciliation of religions or confessions was impossible (Kaplan 2007, p. 31; Kaplan 2014, pp. 1–17; Dumont 2016). Hence, invisible boundaries of "pragmatic toleration" were seen to separate conflicting religions and hostile confessions (François 1991; Christman 2015). It was the philosophers of the Enlightenment that were understood to have developed the concept of tolerance and therefore contributed to the progress of peace, stable political order, and—finally—liberal democracy, by marginalizing religion to the private sphere of personal beliefs (Israel 2001; Zagorin 2003; Şahin 2010; Garrioch 2014; Forst 2013, p. 8).

This vision of the development of state–church relations also had a geographic dimension. As the "European" concepts originated in Western Europe and flourished in America. Eastern Europe, on the other hand, cut off from the philosophy of the Enlightenment, turned from the "paradisus haereticorum" of the sixteenth century to a "haven for intolerance" in the eighteenth century (Butterwick 2001, 2008; Butterwick et al. 2008; Kriegseisen 2016; Schunka 2019; Golebiowska 2014, p. 19). As a result, theses about the separation of Eastern Europe from Western Europe can be found even in studies of seventeenth- and eighteenth-century thought: Wolfgang Schäufele notes that "even where an intra-Protestant concord succeeded in neighboring countries, as in Poland in the consensus of Sendomir (Sandomir, Polish: Sandomierz) in 1570 [...], this remained without effect on Germany" (Schäufele 1998, p. 15).[1]

Despite disagreements over the proper terms with which to describe the modern world and its roots in the early modern era, it is difficult to question this perception as a whole. This article, however, focusing on the irenic initiatives in early modern Europe, attempts to show that other historical currents were coursing through the events of the time and that other solutions to the problem of religious coexistence and religious peace were in play. Howard Louthan described these initiatives as "the efforts of church leaders seeking to minimize doctrinal difference and discover a common theological platform between different Christian traditions" (Louthan 2017, p. 5). In recent historical research, these irenic efforts have remained in the shadow of the notion of a prevailing tolerance. In contrast to tolerance, irenicism is usually depicted as an apolitical movement that shares its genesis with ecumenism, existing on the margins of intellectual life. Against this historiographical vision, the current article presents irenicism as a fundamental idea forged in mainstream intellectual and political life. Focusing on the *Sandomir Consensus*, the paper points out a continuous transfer and exchange of thoughts running in both directions between East and West. The *Consensus* was, thus, both a historical example of successful agreement and a certain model that could be generalized to a different environment. This analysis aims to show that the interpretation of this model changed over the centuries.

The point of departure for this consideration is the origins of the *Sandomir Consensus* as a model of an irenic agreement of early modern Christian confession. Next, the article presents examples of the reception of this agreement in Western Europe, arguing for the great popularity of the document in the seventeenth century among the irenic thinkers and theologians. It concludes by presenting the discussion in the eighteenth century, when the *Consensus* became the foundation of the identity of the Moravian Church emerging in Herrnhut.

## 2. *Sandomir Consensus* as a Model and Example

In the conclusion of his work produced in Heidelberg in 1605, Camerarius published the *Sandomir Consensus*, a document that was sealed in April 1570 in a medium-sized city in the southern part of the Polish–Lithuanian Commonwealth. The agreement crowned long negotiations, conducted at a synod by the Reformed (Calvinists), Lutherans, and Bohemian Brethren. However, it was the Reformed who came up with this idea and organized the synod. It was also the Reformed who drafted the agreement, while the other Protestant denominations remained cautious and reluctant. Finally, after long debates, the Reformed theologians and politicians succeeded in endorsing the document, proclaiming

mutual recognition within the Protestant confessions of the main articles of faith (*de Deo et Sacra Triade*, *incarnatione Filii Dei*, *iustificatione et aliis primariis capitibus fidei Christianae*), an agreement on the Eucharist, and an outline for cooperation. Ultimately, the representatives of all confessions accepted the document (Sipayłło 1972, pp. 301–5; Pelikan 1947; Petkunas 2005, 2009; Ptaszyński 2019).[2]

Historically, the success of the negotiations was surprising. From the early days of the Reformation, the paths of reformers Martin Luther and Ulrich Zwingli shifted ever further apart. From the publication of the *Augsburg Confession* (1530), each denomination had separately codified the main points of doctrine. Lengthy and petty discussions in synods proved fruitless, despite pressure from rulers who were sometimes anxious to create a united confessional front. So it was that at the Marburg Colloquium (1529), Protestant theologians failed to establish a common interpretation of the Eucharist. The synods between Catholics and Protestants in Hagenau (1540), Regensburg (1541, 1546), and Worms (1557) failed to successfully resolve the controversy over the justification and other main articles of faith (Lexutt 1996; Slenczka 2010). Gradually, in the second half of the sixteenth century, discussions showed that differences traversed not only the underlying unity of Christian belief shared by Protestants and Catholics, or the common ground occupied by Protestant confessions, but also the supposedly inviolable internal uniformity of each Protestant denomination. Almost every attempt to impose the unification of religions and silence disputes resulted in the exacerbation of conflicts—not least the Augsburg Interim of 1548, which led to a sharp intra-Lutheran division. In an attempt to impose orthodoxy at the level of the territories of the Holy Roman Empire, rather than on the entire denomination, the secular authorities published compilations of doctrinal texts, called *Corpus Doctrinae* (Dingel 2021, pp. 120–26). Their implementation, however, led to an even stronger fragmentation of the Protestant landscape. The Lutheran *Formula of Concord* (1578) and the *Book of Concord* (1580), intended to quiet the divisions and to unite Lutheranism, turned out to be the *Concordia Controversa* (Dingel 1996).

Therefore, the results of the Sandomierz synod might seem surprising, even if it benefited from past experience. The representatives of three Protestant confessions not only recognized one another's orthodoxies, but also agreed on the main points of doctrine (*primaria capita fidei Christianae*). Last but not least, they established principles of cooperation, including not only joint synods, but also the publication of the *Corpus Doctrinae* containing the three creeds: the Lutheran *Confessio Augustana*, the Reformed *Confessio Helvetica Posterior*, and the *Confessio Fidei* of the Bohemians. Despite many attempts, none of the parties debating in Sandomierz managed to impose its own confession of faith as common theological ground. The secular authority (kings, dukes, or city councils) never confirmed or accepted this agreement.

On the other hand, this *Consensus* was firmly rooted in the tradition of religious dialogue of the sixteenth century, in which councils, synods, and colloquies played a central role (Fuchs 1995; Brockmann 1998). It was clear to theologians that agreements had to be worked out through debate and accepted by those present and then by the rest of the faithful. The apothegmatic phrase "primaria capita fidei Christianae" referred to a construction of irenic humanists and theologians who attempted to bring about doctrinal agreement by distinguishing between articles central to the faith and those that were secondary. While the attention of scholars was traditionally attracted by the concept of "adiaphora," describing it as being in keeping with the Stoic tradition "secondary things" (Friedrich 2007; Spicer 2020), it was, after all, by definition the reverse of the obverse—the establishment of "primary things" or "key points" of the doctrine (Schäufele 1998, pp. 51–53; Schunka 2022, pp. 315–16). In Sandomierz, the "main articles of faith" consisted of the concepts of God, Trinity, Christology, and justification by faith alone. The irenic significance of this construction was obvious: consent on a limited number of fundamental points did not require agreement on numerous secondary issues.

Eventually, regardless of agreement in the main articles of faith, the Protestants in Sandomierz added a note about the sacrament of the Eucharist—the bone of contention

of all Protestant history to date. Protestantism's division over the symbolic and bodily presence of Christ in the Eucharist seemed insurmountable after Luther's and Zwingli's declarations, despite the efforts of John Calvin and Philipp Melanchthon. Yet one successful initiative has been recorded in the history of Protestantism: the Wittenberg Concord, when theologians in the south of the Reich adopted a mildly worded definition of the sacrament of the Eucharist (1536). Citing Irenaeus of Lyon (135–200), the Concord assumed that the sacrament consists of two elements: earthly and heavenly. In the Eucharistic feast, under the forms of bread and wine, the body and blood of Christ are present truly and substantially ("vere et substantialiter"). Transubstantiation is rejected, but it is pointed out that the bread becomes flesh by virtue of the sacramental union ("sacramentali unione"), which does not mean the permanent confinement of the body in the bread ("localiter") but is connected with the act of distributing the sacrament (Bretschneider and Bindseil 1836, vol. 3, pp. 75–77; Bucer 1988, pp. 115–34). Without explicitly referring to the Wittenberg Concord, the authors of the *Consensus of Sandomierz* quoted exactly the same passage from Irenaeus's *Adversus Haereses*.

The mention of the two elements constituting the sacrament, earthly and heavenly, was certainly not among Irenaeus's most important findings. However, its invocation in the Wittenberg Concord, which opened a small window of opportunity for agreement between the cities of the southern Reich and Switzerland, had significance beyond its laconic wording. The document in question, a preface to *Adversus Haereses*, appeared in print in 1526 in an edition prepared by Erasmus of Rotterdam, who died in Basel two months after signing the Concord. In the preface, Erasmus articulated the significance of this edition: "Of all his many works [by Irenaeus] only this present volume has been spared by the envious hand of time"; however, "no one, unless possessed of uncommon patience, will be able to read these books without boredom" (Allen 1926, no 1738, p. 387; Dalzell 2003, p. 295).[3] Complaining of the boredom occasioned by the treatise, the humanist drew attention not only to the church apologist's name, meaning "defender of peace" (*autor pacis*), but also to the readiness for martyrdom present in Irenaeus's writings. A decade later, theologians invoked Irenaeus in Wittenberg as a symbol of religious orthodoxy and the struggle for peace. In Sandomierz, the gesture was repeated, referring to the successful Wittenberg agreement. Analogically, the *Consensus of Sandomierz* was invoked in Heidelberg in 1605, as an annex to the history of the Hussites.

### 3. Irenic Palatinate in the European Network

The small history of the Hussite movement was prepared for print by Ludwig Camerarius (1573–1651), who was then climbing the ladder of a court career in the Palatinate. After long studies at the Protestant universities in Altdorf, Helmstedt, Leipzig, and Basel (1592–1597), the young lawyer joined the ranks of Frederick IV's (d. 1610) and Frederick V's (1596–1632) administration. Soon, he became a councilor (1610), and subsequently a privy councilor (1611), which made him practically the author of Palatinate foreign policy when the elector came of age (Schubert 1955). According to historians, he bore the responsibility for the Palatinate's involvement in the Thirty Years' War and the elector's ascent to his coronation as king of Bohemia (Wolgast 2014).

In the second half of the sixteenth century, and still on the eve of the Thirty Years' War, the court in Heidelberg was one of the most important in the Reich. After the public conversion of the elector and publication of the *Catechism of Heidelberg* (1563), the Palatinate became the first openly Reformed duchy of the German Empire, which was a violation of the *Peace of Augsburg*, allowing as it did only the (Lutheran) *Augsburg Confession* alongside the Catholic one. Following this conversion, Heidelberg became home to the first Reformed university of the Reich, educating Calvinist lawyers as well as clergy (Strohm et al. 2006). The university attracted students from many corners of Europe, where they could not receive an education in the spirit of their confession. One of them was a member of the Church of the Bohemian Brethren, John Amos Comenius (Jan Amos Komensky) who studied in Heidelberg in 1613–1614.

As Christoph Strohm argues, the lawyers not only animated the administration and laid the intellectual groundwork for political reform, but also created a specific Reformed confessional culture, some of whose ideas were taken up by late humanism (Strohm 1996, 2006). Theologians, in turn, made a name for themselves with broad irenic projects, laying the foundation for an agreement between the confessions of Lutheranism and Calvinism (de Jonge 1980; Sarx 2007, pp. 275–83; Selderhuis 2006). On the one hand, the blurring of confessional differences served to legalize and entrench Calvinism in the German Empire. On the other hand, the rapprochement of the denominations facilitated the building of a united front among Protestants in the face of an aggravated political situation. An irenic network tied Protestant Western Europe together.

## 4. The French Connection

As James I ascended the English throne in 1603, he received a congratulatory letter from Jacques-Auguste de Thou (1553–1617). The royal librarian of the Catholic king of France, Henry IV, not only wished him a fruitful reign, but also expressed his hope for success in promoting "the concord of the Church" and sent him the first volume of his *History of his time* focused on the question of religious agreements in Europe (Patterson 1997, pp. 1–3). Needless to say, the *Sandomir Consensus*—as an agreement between Lutherans and Calvinists—found its place on the pages of this historiographic work, when the author reached the year 1570 (de Thou 1734, p. 287).

Undoubtedly, everyone was familiar with the document, as Polish Protestants had informed their western counterparts about the successful synod. Immediately after its enactment in 1570, they not only sent the document to the Helvetic centers of the Reformed theology (Geneva, Zurich, and Basel) but also to some theologians in the German Empire. In 1578, the Polish Protestants sent the *Consensus* to the electors in the Palatinate, Saxony, and Brandenburg, encouraging their efforts at religious peace.[4] The first printed version of the document was—supposedly—published in Heidelberg in 1586 (*Consensus* 1586; Bickerich 1930, p. 363). Between 1605 and 1618, the *Consensus* was printed and quoted several times in the Palatinate.

The document enjoyed popularity among Reformed irenicists, whose number included an author of the Heidelberg Catechism, Zacharias Ursinus (1534–1583), as well as Franciscus Junius (1545–1602) and David Pareus (1548–1622). The irenic theologians had close relations with the Huguenots in France, who had been streaming into Heidelberg since the night of St. Bartholomew. Since their establishment, the College (1559) and then Academy of Sedan (1579) had become important centers for the development of Reformed theology, as did the Academy of Saumur later (1593). In 1604, the future Palatinate elector, Frederick IV, studied in Sedan under the supervision of Daniel Tilenus (1563–1633). Tilenus, together with other leaders of the Huguenots such as Pierre du Moulin (1568–1658) and Philippe Duplessis-Mornay (1549–1623), endorsed the liberal wing of Calvinist theology, seeking to strengthen ties with Protestants both in the Palatinate and in England. Considered the most prominent humanist among Huguenot migrants and also a proponent of irenicism, Isaac Casaubon decided to move to England to the court of James I in 1610 (Callisen 2012). Thus, the congratulatory letter from de Thou, liberal Catholic at the court of a Catholic convert king, was no surprise.

Soon, the ties between the Palatinate, England, and France grew closer, as James I joined the anti-Habsburg coalition that included France, the Protestant Netherlands, and the Union of Evangelical States in Germany. Subsequently (1614), the king's daughter, Elisabeth, married the elector of the Palatinate to become later the "Winter Queen" of Bohemia, when Frederick was crowned by the Bohemian nobles in 1619, which began the Thirty Years' War. Frederick also constantly maintained close relations with the Huguenots in France. It was probably in cooperation with the king that the National Synod of Huguenots in Tonneins prepared guidelines for a Protestant agreement (Patterson 1972; Maag 2018). As an exemplar of the successful unification of the Protestants, the *Sandomir Consensus* stood out (Aymon 1710, vol. 2, p. 61).

**5. Irenicum (1614) and *Historica Narratio* (Camerarius 1605)**

Among the irenic works produced in Heidelberg, David Pareus's *Irenicum* (1614) certainly played the most important role (Selderhuis 2006). According to the theologian, agreement could only be established among Christians through a universal synod (council), which would propose a common foundation for the faith through discussion (Pareus 1614). Therefore, the deliberations required particular preparation—starting with the selection of participants, defining the topics, and determining the place and time. The subject of discussion should be the main articles of faith (*articuli fundamentales*). Among the examples of successful agreements joining the two confessions, the theologian enumerated three of the most important: the Marburg Colloquy, the *Wittenberg Concord*, and the *Sandomir Consensus*.

Pareus not only mentioned the *Consensus*, but also published the entire document (chp. 22) along with letters sent in 1578 to the elector of the Palatinate (chp. 23). Even though he mentioned the three denominations that signed the agreement, the theologian made no secret of the fact that its most significant aspect was—at least in his eyes, and despite appearances—the rapprochement between Lutherans and Calvinists in agreeing on the most important point, distinguishing them from the Catholics (Pareus 1614, p. 135).

The historical work edited by Ludwig Camerarius employed a different method, still citing the *Consensus*. The main body of the text was entitled *Historica Narratio* and told the history of the Bohemian Brethren. The author of *Narratio* emphasized that Bohemians should be distinguished from the Picards and Waldensians (*Historica Narratio*, Camerarius 1605, pp. 6–12, 59). It was a group of Jan Hus's supporters, who were reading the works of John Wycliffe, protesting the abuses of the late medieval Church, who finally formed the Bohemian Brethren. After Hus's death at Constance (1415), his followers were initially all called Hussites in their entirety, although the party soon splintered into several groups. Among them, the moderate Calixtines (called also Utraquists) and the radical Taborites were the most significant. The author leaves the reader in no doubt that the Brethren developed from a group dissatisfied with the rapprochement between the Utraquists and the papacy, thus continuing the tradition of radical faction (*Historica Narratio*, Camerarius 1605, pp. 80–91). Their main concern was with the purity of Christian life, not the dogma that remained close to the Roman church: "Above all, they did not want to preserve the pure doctrine and teaching but a discipline, which was essential for Christian piety" (*Historica Narratio*, Camerarius 1605, p. 92).[5] It did not take long before a group of independent individuals had organized themselves into an institutional framework. In 1457, the Brethren formed the first community and, ten years later, selected by lot representatives, who subsequently received an ordination from the Waldensian bishop. After 1517, the Brethren contacted Martin Luther and reformers from the south of the Reich and Switzerland. Finally, in the middle of the century, they came to Poland, where they held talks with John à Lasco. To this treatise written by Joachim Camerarius, Ludwig Camerarius appended thirteen sources illustrating the history of the Brethren. The last among them was the *Sandomir Consensus*.

From the perspective of modern research, the story told by Camerarius is not free from errors and simplifications. For example, the apostolic succession received from the bishop of Waldenses is regarded as a legend (Atwood 2021, fn. 10). However, another observation is essential for this argument. The logic of the historiographical work edited by Ludwig Camerarius was quite different from that of Pareus's *Irenicum*. *Historica Narratio* placed the *Consensus* in the context of the history of Hussitism, not the dialogue between Lutherans and Calvinists. Hussitism and its evolution in the form of the Bohemian Brethren provided an answer to the fundamental question of the anti-Reformation polemics: "Where was your Church before Luther?" For Calvinists, this question was particularly uncomfortable, for it not only pointed to the relatively young historical metrics of their confession but also referred to the figure of Luther, with whom they could not always identify. Even as Calvin forged friendly ties with Melanchthon (Frank and Selderhuis 2005), and the Heidelberg Irenicists emphasized that they recognized the *Confessio Augustana variata*, the question of historical origins remained sore. For their part, the figures of Hus and his heirs drew

attention to antipapal statements in the late Middle Ages and criticism of the Roman Curia and church hierarchy (Haberkern 2016).

Why, then, did Ludwig Camerarius use the *Consensus* to accompany the history of Hussitism? Why did he not simply place the irenic document in the context of the Polish Reformation and the history of Calvinism, as Pareus did? Apart from the political situations in the respective polities, the growing conflict between the Habsburgs and the Bohemian estates, and Camerarius's plan to link the Reich with Bohemia, the identity of the Brethren could serve as a suitable answer. As indicated, the core of the Brethren's identity was not any particular doctrine that could differentiate them from other confessions, but a commitment to discipline and a rigorous lifestyle (*Historica Narratio*, Camerarius 1605, p. 142). Emphasizing the role of discipline and simplicity of life was undoubtedly a reference to the book title turned slogan, *Reformatio vitae*, popular in the early seventeenth century and signifying an identification with the Reformed Church. The calls for *Reformatio vitae* or for a second reformation (*Nadere Reformatie*) were demands to perpetuate and revitalize the Reformation, initiated by Luther (and the early Reformers), but in the field of doctrine, not lifestyle. For Calvinist theologians, it was both an apology for their own activities and a chance to secure the recognition and legalization of the Reformed Churches in the German Empire. At the same time, it was a response to voices of complaint about the decline of piety. These complaints found expression in Lutheran pietism (Schunka 2022; Breul and Hahn-Bruckart 2021). However, members of the other Christian confessions had articulated similar concerns. Put briefly, the Brethren offered answers to many of the ills of Camerarius's times.

From this perspective, the Brethren seemed to be a perfect answer to the needs of Heidelberg. Their genesis lay in pre-Reformation times, when the Roman Curia had also recognized their existence, and in the sixteenth century they gained the approval of all orthodox Protestant Churches. In addition, their position on doctrinal issues made religious compromise feasible. Finally, the concept of a community of Brethren as a unity and not an institutional church held the unstated promise of new forms in which to reconcile living faith, Christian discipline, and institutional order. What Pareus expressed with a complex argument at the level of abstract reflection, Camerarius articulated through a historical example.

### 6. "We Are All Hussites"—Confessional Past of the *Consensus*

Having said all this, we must still bear in mind that *Historica Narratio* was not the work of Ludwig Camerarius, as the advisor to the elector of the Palatinate reported in a dedicatory letter to Maurice, Landgrave of Hesse-Kassel (1572–1632). According to Ludwig, the work had been written by his ancestor, the great humanist Joachim Camerarius (1500–1574), twenty (in fact at least thirty) years earlier. As he argued, since research on Occitan language had blossomed thanks to Joseph Justus Scaliger, the churches of Bohemia and Moravia also deserved to be remembered (*Historica Narratio*, Camerarius 1605, **4v–5r). The invocation of Scaliger, a Reformed scholar regarded as the greatest mind of the early seventeenth century, was probably a deliberate reference to the scholarly passions of the Hesse ruler, known as "the Learned." Just as Philipp of Hesse (1504–1567) had been the patron of Joachim Camerarius at the time of the Reformation, Maurice the Learned might become the patron of Ludwig Camerarius.

About thirty years earlier, when Joachim Camerarius was working on the manuscript, the choice of the Brethren as the subject of a historical work had a different meaning. It reflected the interest in the fate of Hus and the Hussites that had developed in relation to Luther. However, it was Luther's opponents who first invoked Hus and Hussitism to criticize nascent Lutheranism. On the eve of the Reformation, the reasons for selecting such an argument were obvious. Luther's opponents intended to show that the new criticism was merely a repetition of heresies that the Roman Church had already defeated in the past. However, the nickname "Lutherus bohemicus" had a political cast as well. It reminded the audience all too well of the Hussite Wars that had destroyed Bohemia.

Therefore, the accusation of Hussitism was a convenient weapon, as it argued that undermining religious dogma could lead to social conflicts and even wars (Haberkern 2016; Bagchi 1991, pp. 69–92; Edwards 1990).

The loudest accusations against Luther were formulated by Johannes Eck (1486–1543) at a colloquium in Leipzig in 1519 (Leppin and Mattox 2019; Wurm 2011). As is well known, these accusations surprised Luther, who had to admit that he considered many of Jan Hus's theses to be correct. Soon, he also wrote a famous sentence: "Sumus omnes Hussitae ignorantes" ("We are all Hussites without knowing it") (Hendrix 1974; Oberman 1997). Subsequently, the claim of repeating Hus's heretical statements was eagerly formulated by both the Catholic controversialists (Johannes Cochläus, Johannes Fabri) and Lutheran theologians, and thus became a part of Protestant confessional identity. Symbolically embodying this transformation were images of Luther with a swan and a goose, increasingly popular after 1530 (Fudge 2016, pp. 254–76; Kolb 1987, p. 146ff). As a result, Hus entered the canon of witnesses to the truth (*testes veritatis*), codified in the great historiographical project of the *Magdeburg Centuries*. The initiator and one of the authors of the *Centuries* was Matthias Flacius Illyricus, who is now also considered one of the main authors of Lutheran orthodoxy. When it became clear that the paths of the *Centuries'* authors were going to diverge in the work's fifteenth *Century* (never published), even before some of the source documents from the 1400s had been published, Flacius independently published a volume on Hus (Flacius 1558). Almost simultaneously, Flacius printed in Basel a smaller treatise: *The catalogue of the Witnesses of the Truth* (Flacius 1556; Scheible 1996).

Flacius's argument in that document was a critique of the institution of the papacy. Right from the start, Flacius argued extensively that the pope was the Antichrist, against whom the witnesses to the truth were acting. In the unbroken chain of critics of the pope calling him the Antichrist, Flacius assigned an important place to Hus and Jerome of Prague. Passionately, Flacius quoted Hus's prophetic remarks that the swan would come after the goose and that a hundred years after Hus, a reformer would come whom the Church would not be able to silence. However, Flacius intended his portrayal of the Bohemian Churches to be taken not without a grain of salt. He depicted the Bohemian Churches as internally divided and fractured, drawing a parallel with the situation of Lutherans in the Reich. What is more, making extensive use of Catholic historiography, he identified the Hussite Churches with the Waldenses, who seemed to him a much more interesting group. Flacius devoted a long chapter to the Waldenses, focusing on their medieval origins and explaining that—contrary to popular opinion—the movement's supporters were not representatives of the lower social strata but of aristocrats and magnates. As a result of divisions and very violent persecutions, the Waldensians scattered across France and throughout Europe. This observation prompted Flacius to mistake many references to the Hussites he found in older historical works as references to the Waldensians.

The history penned by Joachim Camerarius was an obvious polemic against the vision of Flacius, on the one hand. On the other hand, it was a part of a debate about the identity of the Lutheran confessional culture. At this moment, Camerarius aligned himself with the most prominent Greek scholars of the sixteenth century but was also a friend of Philipp Melanchthon. After Melanchthon's death, Camerarius published his first biography (1566) and a collection of correspondence (1569). Behind both works was an undisguised desire to defend Melanchthon from attacks that fell on him for—among other things—taking irenic initiatives (Woitkowitz 1997).

As noted above, it was the identification of the Bohemians with the Waldenses that provoked the liveliest protest from Camerarius (despite the obvious historical ties linking the two churches). Camerarius critically and even maliciously notes that Flacius confused people and mixed up the chronology because he was a man more apt to quarrel than to study and reflect (*Historica Narratio*, Camerarius 1605, p. 52). Moreover, Camerarius presented the social profile of the Bohemian church differently. In contrast to the elitism of Flacius, Camerarius made no secret of the fact that the Brethren originated from a radical faction of Hussitism suspected of inciting armed conflict. The Brethren were not members

of the aristocracy or nobility, although in Bohemia or Poland they often enjoyed noble protection and patronage. The importance of this passage is highlighted by the praise of the Brethren at the end of the treatise: the brothers had no ambition, did not involve themselves in doctrinal disputes, and were not driven by greed (*Historica Narratio*, Camerarius 1605, p. 142). These characteristics distinguished the Brethren from other confessional groups. In particular, their openness distinguished the Brethren from the Gnesio-Lutherans, who criticized Melanchthon for trying to make compromises in confessional matters. Contrary to their opponents, Melanchthon, Camerarius, and their supporters saw the Brethren as a model of open Christianity.

### 7. Political Future of the *Sandomir Consensus*

The tragedy of the Thirty Years' War and the limited success of the *Peace of Westphalia* did not mark the end of the popularity of irenic projects, including the *Consensus of Sandomierz*. On the contrary, it stepped back into the spotlight during the Colloquium Charitativum (1645), organized in Thorn at the end of the war (Müller 2004, p. 274). Seeking a compromise agreement, many thinkers plotting peace projects evoked the *Consensus* in their works and correspondence. However, the court of the electors of Brandenburg (soon the kings of Prussia) in Berlin was especially interested in this project. The situation in Brandenburg became extraordinary when the elector converted to Calvinism in 1613, while allowing the nobles and citizens to retain their Lutheran confession. The conflict soon became muddled on three fronts: Protestants vs. Catholics, Lutherans vs. Reformed (Calvinists), and Pietists vs. Lutheran orthodoxy. A fourth front, recently highlighted by Alexander Schunka, was the clash among Reformed theologians, who also failed to form a unified camp (Schunka 2019, pp. 29–30).

As Schunka elaborated, the Calvinist Hohenzollern court often served as a bridge between Western and Eastern Europe, inspired by the English irenicism and involved as it was in Polish–Lithuanian politics. For the next two hundred years, Brandenburg's *raison d'état* or even *raison d'être* was to preserve the unstable balance between the disparate confessions, while at the same time, the ruling dynasty of Hohenzollerns sought to exploit the Reformed confession in their policy. As Alexander Schunka put it, "Irenics thus served the *raison d'état*; it was implemented by Reformed personnel such as the court preachers, whose theological basis was universalistic Calvinism" (Schunka 2019, p. 40).[6] The most famous expression of this policy was the *Edict of Potsdam* of 1685, guaranteeing the admission of Huguenots expelled from France.

Court preachers, active at the court in Berlin and Königsberg, were an important tool of this policy. On the one hand, they strove to promote Calvinism, while, on the other hand, they participated on behalf of the elector in many irenic initiatives undertaken in Europe, among them the Colloquies of Leipzig (1631), and the Colloquium Charitativum in Thorn (1645), in Kassel (1661), and in Berlin (1662/63) (Ruschke 2012). In 1703, the king of Prussia organized a Collegium Charitativum in Berlin, which Wolf-Friedrich Schäufele described as "the last religious discussions on the inter-Protestant union in Germany" (Schäufele 1998, p. 22)[7]. When making appointments to the post of court preachers in Königsberg, the electors reached out to clergy from the liberal wings of Reformed Churches, often those with ties to the Brethren. So it was that, in 1643/44, they appointed Bartholomäus Stosch (1604–1686), who was born in Silesia to a family of Dutch emigrants, had been ordained in the Polish city of Leszno (Lissa) by Bohemian Brethren, and—as a court preacher in Königsberg—had edited the records of the Colloquium Charitativum. After his death, he was succeeded by Daniel Ernst Jablonski (1660–1741). The preacher originated from the Brethren, was educated in Oxford (1680–1683), and while in Prussia, he remained a member of the Brethren and its bishop ("senior"). He maintained correspondence ties with Poland, helped Polish and Lithuanian students sent to Brandenburg–Prussia, and, when allowed by the elector, he visited the synods of the Protestant Churches in Poland. The *Sandomir Consensus* was also considered a part of the Brethren's legacy (Bahlcke and Korthaase 2008; Schunka 2019).

In 1704, a theology professor in Frankfurt an der Oder, Samuel Strimesius (1648–1730), edited the *Sandomir Consensus* and translated the document into German (Strimesius 1704). In his introduction, the editor situated the *Consensus* in the context of the religious colloquies in the German Empire in Marburg (1529), Wittenberg (1536), Leipzig (1631), and Kassel (1661). In contrast to these famous attempts, the almost forgotten *Consensus* was a successful project, accepted and brought to life by the participants. According to Strimesius, the *Consensus* gained particular significance on account of the situation in Poland, where—on the one hand—Protestants became a persecuted minority, and—on the other—the war with Lutheran Sweden was just about to begin.

A remark by Strimesius about the political situation made reference to the rivalry between the electors of Brandenburg and Saxony. The object of this rivalry was initially a place in the confessional mosaic of the Reich and leadership in the Corpus Evangelicorum, established by the Peace of Osnabrück and which, since 1653, brought together Protestant territories. The director of the Corpus Evangelicorum was the elector of Saxony, which became problematic after 1697, when he decided to reach for the Polish throne, and converted to Catholicism. Subsequently, political ambitions led the king of Poland to join an alliance with Denmark and Moscow, against Sweden, which unavoidably led to the Great Northern War (1700–1721). At the beginning of the war, Brandenburg–Prussia maintained an ambivalent position, holding talks with all sides. Soon, the elector of Brandenburg seized the opportunity to crown himself king in Prussia. In 1703, the king of Prussia signed a treaty with the king of Sweden, providing for the protection of Protestants in Poland (Hassinger 1953, pp. 87–121). Strimesius published the *Consensus* when, a year later, military defeats forced King August II to leave Poland and subsequently to abdicate.

Soon, Strimesius's numerous irenic treatises found attentive readers among members of the Corpus Evangelicorum, who debated the union of Protestants in Regensburg in 1719–1722, with the cooperation of theologians from Prussia, Hannover, and Württemberg at the root of the discussion. After Leibniz's death (1716), Christoph Matthäus Pfaff (1686–1760), Daniel Ernst Jablonski, and Archbishop of Canterbury William Wake (1657–1737) played first fiddle in this group, which included Jean-Alphonse Turrettini (1671–1737) and Johann Christian Klemm (1688–1754). Pfaff, a lecturer in Tübingen who made irenicism his life's work, also made church history a fundamental point of reference. His fascination with church history and ambition pushed him into forgery. The erudite young man published his writings as a supposedly newly discovered work by Irenaeus of Lyon (Schäufele 1998; Edsall 2019). For Jablonski, who not only received his education at Oxford but also had close relationships with Anglican theologians and planned a union with the Anglican Church, the talks presented another opportunity to pursue irenic projects (Schunka 2008; Nishikawa 2008; Schunka 2019). As a result, dozens of short irenic treatises appeared in print and were subsequently commented on in correspondence networks. Along with the Marburg Colloquy and the Wittenberg Concord, the *Sandomir Consensus* was omnipresent (see Turrettini 1707, p. 22; Turrettini 1719, pp. 162–65; Pfaff 1720, pp. 6–7).

As usual, the irenic projects triggered a wave of polemics in the Reich. Numerous Lutheran theologians, among whom Ernst Salomon Cyprian (1673–1745) was the most prominent, spoke out loudly against the project. Turning on the Consensus, a Hamburg Lutheran pastor, Johann Theodor Heinson (1666–1726), argued that Lutherans had only accepted the *Sandomir Consensus* because the treacherous Calvinists had deceived them. The compromise formulation of the Eucharist in the *Consensus* went against Lutheran tradition. As a result, the *Consensus* had enabled the Reformed communities to take over the Lutheran churches, which had almost disappeared from the map of Poland (Heinson 1721, pp. 93–95).

Responding to their opponents, supporters of Protestant union also frequently (and just as tendentiously) cited the example of the *Sandomir Consensus*. Johann Christian Klemm claimed that the *Consensus* not only laid a solid foundation for the cooperation of "evangelical" churches but was also approved by the kings of Poland–Lithuania (Klemm 1724, pp. 95–97). His use of the term "evangelical" was significant. Klemm, Pfaff, and some

other irenicists advocated for dropping the terms "Lutherans" and "Calvinists" in favor of "Evangelicals" (Schäufele 1998, pp. 207, 256; Schunka 2022). Using this term to describe the situation in Poland, Klemm suggested that in 1570, the Lutherans and Reformed had established a single Protestant church, which was legalized by the monarchs. None of these claims was true.

## 8. Reinvention of the *Consensus*

The circle of correspondents discussing irenicism and the unification of Protestant confessions included Count Nikolaus Ludwig von Zinzendorf (1700–1760) of Saxony (Schäufele 1998, pp. 189–92; Daniel 2004). On 15 July 1729, Count Zinzendorf wrote a letter to Daniel Ernst Jablonski, at the time a court preacher in Berlin. The count reported that refugees from Bohemia were settling in his lands and asked the court preacher for a "letter of encouragement" to the congregation, which consisted of successors to the old Bohemian refugees.[8] After barely four weeks, the court preacher replied with an extensive explanation of his confessional standpoint and the history of the origin of the Brethren congregation in Poland–Lithuania. Jablonski clarified that the "little group of Bohemian and Moravian Brethren [were] the forerunners and morning light of the Reformation." After the Thirty Years' War, persecutions in Bohemia and Moravia had forced them to emigrate to Poland–Lithuania, where the Brethren "have survived until now." In Poland, they belonged to the "Reformed Christians," but kept their traditions. "They profess their confession, call themselves the Bohemian Brethren, and use their discipline and church order." The personal fate of Jablonski was also part of this migration: "My forefathers were born in the same church, and I was born in exile in Poland, I grew up in the same church, and I imbibed the love for the same church with my mother's milk. Although God tore me away from them during my lifetime, and transferred me to this country, the king [of Prussia], who rested in God, and the king, who is now reigning [in Prussia], have all but decided that I should co-administer the episcopate of this church".[9]

Jablonski, a grandson of Comenius (Komensky), served simultaneously as a court preacher in Berlin and as a bishop of the Bohemian Brethren in Poland. As Alexander Schunka has argued, he was also a "missing link" between the Bohemian Brethren in Poland and the Moravian Church established by the wealthy Lutheran count in Saxony (Schunka 2010, p. 56). However, Zinzendorf's interest in the Brethren did not stem from home or university. During his education, which also included a few years at the Pietist Pedagogical College in Halle (1710–1715), he came under the influence of the Lutheran pietism and dominant personality of August Hermann Francke. After studying in Wittenberg (1716–1719), he entered the service of the king of Poland and elector of Saxony (1721–1732). In 1722, he acquired his grandmother's estate in Lusatia, on the Bohemian, Silesian, and Saxon border, called Berthelsdorf. Soon, refugees from Bohemia and Moravia settling in his estate exposed the Lutheran nobleman to new challenges (Peucker 2022). While his pietist background and the strong influences of some other radical currents of the time (like the Philadelphian movement) made him highly critical of the orthodox confessional cultures, the Bohemian migration forced him to look for a compromise. What is more, the Bohemian settlers—even if they belonged to different denominations (such as Utraquists, Lutherans, or Catholics)—made use of Bohemian songbooks or the Comenius Catechism. Thanks to them, Zinzendorf became acquainted with the heritage of the Bohemian Brethren. This was the reason he turned to Jablonski in 1729, and eight years later, he became a bishop of the Brethren.

The ordination of Zinzendorf in 1737 was a pragmatic decision by the two theologians, preceded by a long hesitation. Jablonski had to reckon with the political calculations of the king of Prussia, while at the same time nourishing hopes for the union of the Protestant Churches (Schunka 2019). Zinzendorf first turned to the king of Denmark in 1731, seeking to legalize church structures in his territories. Then in 1735, he sent his colleague, David Nitschman to Jablonski and the Brethren for ordination.

The connection with the Brethren enabled Zinzendorf to establish the stability of their church structure and to undertake missionary activities. In addition, the notion of the apostolic succession, to which the Brothers owed their recognition by the Church of England in 1717, may have seemed attractive. However, as Alexander Schunka has pointed out, Count Zinzendorf appreciated neither the dignity of the bishopric nor the history of the Brethren. He also understood the idea of uniting the churches differently from Jablonski: not as a union of churches, but as a mutual tolerance between the churches, based on the community of the true invisible Church (Schäufele 1998, p. 192; Daniel 2004; Schunka 2010, pp. 62–64).

Was it possible, then, that the *Consensus* was a reason why Zinzendorf took an interest in the Brethren? A few years before Zinzendorf's ordination, in 1731, Jablonski had published the history of the *Sandomir Consensus* (Jablonski 1731). Immediately, he sent an exemplar of the book to Zinzendorf. Subsequently, the confessional agreement appeared in the scenes that Zinzendorf ordered from German–American painter Johann Valentin Haidt to decorate his house in London (Atwood 2013, p. 115; Schunka 2010). Even if Zinzendorf claimed that the Moravian Church fell under the umbrella of the Lutheran communities, the first historian of the Moravian Church, David Cranz (1723–1777), placed the *Consensus* as one of the main elements in his work, depicting the *Consensus* as the link between the two communities, Bohemian and Moravian (Noller 2016). As Craig Atwood cogently put it: "*The Consensus of Sandomierz* excited Zinzendorf's imagination and offered a way forward for his Philadelphian Brüdergemeine. [...] Zinzendorf was willing to tolerate the Moravians in part because he believed that the Brethren had always been ecumenical in outlook" (Atwood 2013, p. 115). First there was the *Consensus* as history, as told by Lutherans, Calvinists, and the Brethren over the centuries, and only later do we see the *Consensus* as instrument, a living tool with which to renew the community.

## 9. Conclusions

If the motivation for the renewal of the Bohemian Brethren was indeed the belief that the *Consensus* was authored by the Brethren, there could hardly be a greater mistake. In 1570, the Polish branch of the Bohemian Brethren simply recognized and accepted the model of agreement designed by the Calvinists.

However, Zinzendorf would not have been entirely wrong. A look at the Brethren reveals the evolution of the image of this community, from being at first sometimes perceived as a primitive Hussite church to gaining a reputation as an openly independent religious community, and eventually as a Reformed church. The *Consensus* played a pivotal role in this evolution of image and identity. In Poland, the *Consensus* led to a gradual rapprochement between the Brethren and Calvinists, with the Brethren subsequently adopting the identity of the Reformed Church. At the end of the sixteenth and seventeenth centuries, in the eyes of many theologians and irenic thinkers, it was an example of a successful agreement reached through the initiative of an "independent third," a community that broke down the Lutheran–Calvinist dichotomy. The Brethren have become an alternative to confessional divisions, and the *Consensus* has become part of their heritage.

Looking at this evolution, it is difficult not to notice the role of the political factor in the emergence of the document, as well as in changing the meaning of the document. Before the Thirty Years' War, the burning need of Protestants was to form a united front against the Catholic Union. In France, it was important to create a political community after the years-long religious war, only just ended by the *Edict of Nantes*. In England, too, James I needed to find common ground with Anglican Calvinists, Presbyterians, and the Catholic minority. Finally, in Brandenburg–Prussia, the *Consensus* promised the unification of Calvinists and Lutherans, and allowed for the taking of political steps against the king of Poland and the elector of Saxony.

The long-lasting popularity of the *Consensus* proves its usefulness in shaping the relationship not only between the confessions but also between the state and the churches. Contrary to the belief that peace could only be achieved through the separation of state and church and the recognition of religion as a matter of private beliefs, the *Consensus* is

an example of an alternative solution that enabled confessional coexistence in the public, political domain. In other words, it was not only the Enlightenment and secularization that ended the religious wars but also the continued efforts of theologians who developed new models of religious coexistence, drawing on historical experience. As numerous studies confirm, this coexistence was the rule rather than the exception in the early modern era. Moreover, the constant redefinition of the *Consensus* makes us aware of the actuality of this model, which (despite its historical nature) was more than just an artifact or rhetorical embellishment. Finally, the historical placement of the *Consensus* in Eastern Europe is also evidence of the permanence of intellectual heritage and constant exchange. If evidence of this process is needed, the Moravian Church surely provides it, through which the *Consensus* tradition also found its way to America.

**Funding:** This research was funded by National Science Center, Poland, grant number [2018/31/B/HS3/00351].

**Institutional Review Board Statement:** Not applicable.

**Informed Consent Statement:** Not applicable.

**Data Availability Statement:** Not applicable.

**Conflicts of Interest:** The author declares no conflict of interest.

## Notes

[1] "Auch wo im angrenzenden Ausland eine innerprotestantische Konkordie gelang, wie in Polen im Konsens von Sendomir (Sandomir, poln. Sandomierz) 1570 [...] blieb dies ohne Auswirkung auf Deutschland".

[2] Národní muzeum (Prague), Fragm. 1 E b 1/3, fol. 1r–38v.

[3] "Ex his tam multis viri lucubrationibus solum hoc quod nunc damus, seculorum invidia reliquum esse voluit [...]. quos nemo nisi patientis stomachi poterit absque tedio pervolvere".

[4] Geheimes Staatsarchiv Preußischer Kulturbesitzt, HA I, Rep. 7, Preußen, no. 69, fol. 612, 619.

[5] "Magnopere cupiebant non solum doctrinam puram et integram retinere, sed conservare etiam per omnia disciplinam, et exequi caetera, quae Christiana pietas requirit".

[6] "Irenik diente mithin der Staatsräson; implementiert wurde sie durch reformiertes Personal wie die Hofprediger, deren theologische Basis ein universalistisches Reformiertentum war".

[7] "Die Verhandlung des "Collegium charitativum" in Berlin waren die letzten Religionsgespräche zur innerprotestantischen Unionsfrage in Deutschland".

[8] Unitätsarchiv Herrnhut, sign. R 4 D 1, Nr. 8, Zinzendorf to Daniel Ernst Jablonski, Herrnhut 15 July 1729.

[9] Unitätsarchiv Herrnhut, sign. R 4 D 1, Nr. 13, Jablonski to Zinzendorf, Berlin 13 August 1729.

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
