# Peer review of "Was a Confessional Agreement in Early Modern Europe Possible? On the Role of the Sandomir Consensus in the European Debates"

_religions, doi:10.3390/rel13100994_

Round 1

Reviewer 1 Report

This essay is a truly fine article--well-written, well-argued, well-researched. It holds together quite well, and the author follows the road map he lays out in the beginning throughout the article. It is as good an article as I've ever read. I commend the author.

The only thing I would change, and it is extremely minor and a typographical error, is at line 119, where there is a phrase that is doubled (the one on difference). 

Author Response

Dear Reviewer, Thank you very much for your kind words and encouraging comments. I take this as an encouragement and motivation for further research. I have corrected the error in line 119.

Reviewer 2 Report

This is a valuable contribution to the study of religious toleration in the early modern period that incorporates research on political and religious history. The author is engaged in the most up to date research while showing deep familiarity with the primary sources. The argument is persuasive. I have just a couple of suggestions for refinement.

1. It was Frederick V who married the daughter of King James I (Elisabeth) rather than James marrying Frederick's daughter. This needs to be corrected, but the basic idea of connections with England, the Palatine, and France is good.

2. The history of the Bohemian Brethren that were published (and discussed) are not always accurate. It should be noted that the story of a Waldensian "bishop" is legendary and has been debunked for decades. It would be helpful to know if this supposed "apostolic succession" was important to Flacius and Cammarerius or if they were just reporting what their sources said. It was important to Jablonski in his efforts to unite the Lutherans and Anglicans.

3. Since the author notes the role of England in this history, some mention of Jablonski's travel to England might illuminate further the scope of Prussian ambition related to irenicism. 

4. The author should at least note that Comenius was a student in Heidelberg while Pareus was writing his Irenicum. It seems possible (likely) that the misattribution of the Brethren as authors of the Consensus may have been encouraged by Comenius. 

5. Would it be helpful to briefly contrast the Consensus with the equally famous Confessio Bohemica that was written by Lutherans in Prague but which was signed by the Brethren and Reformed? 

6. In discussing the political situation, it might be helpful to note that Frederick V was crowned by the Bohemian nobles as king in 1619, which began the 30 Years War. In other words, there was a lot of contact between Heidelberg and Bohemia in the first two decades of the 17th century. 

7. The suggestions about Zinzendorf and the Consensus are appropriately nuanced and well-founded. The author mentions the differences between Zinzendorf and Jablonski's understandings of ecumenism but does not explore them further. The missing component is the Philadelphian movement, but it is not necessary to go into that since the thrust of the article is on the 17th century.

8. If I understood the main argument correctly, this lively interest in the Bohemian Brethren and the Consensus demonstrates that in the Reich there were theologians/pastors who were advocating for religious toleration or perhaps Protestant union, which significantly challenges the prevailing opinion that it was the Aufklarung and secularization that ended religious warfare and persecution. That should perhaps be more clear in the conclusion (if I read correctly). 

9. I have not read the Camerarius history, but am I right in assuming that it does not include the contacts between the Brethren and Reformed theologians in Strasbourg and Geneva that may have significantly influenced the theology of Bucer and Capito who had sought reconciliation with the Lutherans? It is possible that this was not known in the early 17th century, but it would certainly add to the misunderstanding that the Brethren were responsible for the Consensus. 

Author Response

Dear Reviewer,

Thank you very much for your kind words and your encouraging comment. I  appreciate you taking the time and effort to read the article. It is an invaluable help. I have done my best to make all the changes you suggested and to address your concerns:

  1. thank you very much for catching my stumble on Frederick's marriage (lines 266-8).
  2. I added a sentence regarding Camerarius' mistakes and the credibility of his narrative (lines 317-319). It seems to me that for Flacius and Camerarius, apostolic succession was not an important part of history. However, it is certainly an issue worthy of a separate study.
  3. Jablonski - I added a reference to his fascination with England and Anglicanism (lines 540-542). Alexander Schunka elaborated extensively on this issue, and I do not want to duplicate his findings here.
  4.  Comenius in Heidelberg - I added a reference (lines 216-219). The hypothesis you propose about Comenius' influence on Pareus is very intriguing. Unfortunately, I do not have sources to confirm it.
  5. The Confessio Bohemiaca is often cited alongside the Consensus Sendomiriensis in the irenic treatises. In my eyes, this is about agreements of a different kind: the tactical arrangement of a confession of faith (close to Lutheranism) to submit it to the ruler for approval. I would prefer not to develop this thread here.
  1. Frederick V and the beginning of the war. This is a key issue - I had mentioned it briefly (line 206), and now added another brief mention (line 268).
  2. the role of the Philadelphian movement - is mentioned in the quote from Craig Atwood (line 636).
  3. I made a minor correction in the conclusions to bring out the thought more clearly (lines 671-673)
  4. Indeed, Camerarius does not write about the Brothers' conversations with Bucer or Capito.

Reviewer 3 Report

I have not done research on irenicism and specifically Sandomir, but I have read a good deal about it and know the general field of Lutheran-Calvinist relationships quite well.  This is a well-crafted, insightful analysis that will be of great benefit.  I found no places where I could make suggestions. and know of no other bibliography that might be consulted.

Author Response

Dear Reviewer,

thank you very much for your kind words and your encouraging comment. I take it as encouragement and motivation for further study.